# MMDRFuse: Distilled Mini-Model with Dynamic Refresh for Multi-Modality Image Fusion

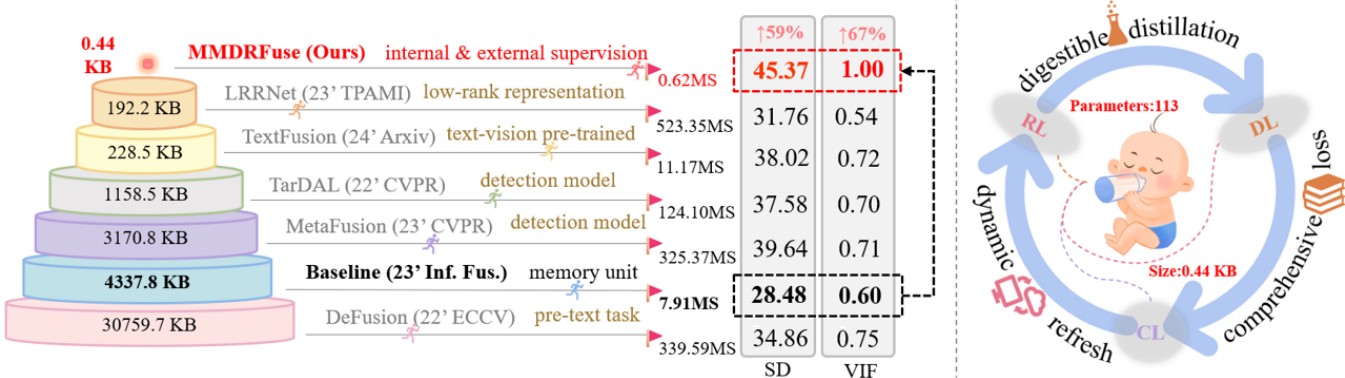

**Figure 1: Left: Comparison of our Mini Model (MMDRFuse) with other advanced image fusion solutions in terms of model efficiency, including model size (KB), average running time (MS), and two image quality assessments SSIM and VIF. Right: Supervision designs of our MMDRFuse, where digestible distillation refers to delivering external soft supervision, dynamic refresh emphasises the historical impact of network parameters during training, and comprehensive loss serves as an internal supervision that absorbs nutrients from source images.**

## ABSTRACT

In recent years, Multi-Modality Image Fusion (MMIF) has been applied to many fields, which has attracted many scholars to endeavour to improve the fusion performance. However, the prevailing focus has predominantly been on the architecture design, rather than the training strategies. As a low-level vision task, image fusion is supposed to quickly deliver output images for observing and supporting downstream tasks. Thus, superfluous computational and storage overheads should be avoided. In this work, a lightweight Distilled Mini-Model with a Dynamic Refresh strategy (MMDRFuse) is proposed to achieve this objective. To pursue model parsimony, an extremely small convolutional network with a total of 113 trainable parameters (0.44 KB) is obtained by three carefully designed supervisions. First, digestible distillation is constructed by emphasising external spatial feature consistency, delivering soft supervision with balanced details and saliency for the target network. Second, we develop a comprehensive loss to balance the pixel, gradient, and perception clues from the source images. Third, an innovative dynamic refresh training strategy is used to collaborate history parameters and current supervision during training, together with an adaptive adjust function to optimise the fusion

network. Extensive experiments on several public datasets demonstrate that our method exhibits promising advantages in terms of model efficiency and complexity, with superior performance in multiple image fusion tasks and downstream pedestrian detection application.

## CCS CONCEPTS

• **Do Not Use This Code → Generate the Correct Terms for Your Paper**; *Generate the Correct Terms for Your Paper*; Generate the Correct Terms for Your Paper; Generate the Correct Terms for Your Paper.

## KEYWORDS

multi-modality image fusion, feature-level distillation, dynamic refresh, end-to-end training

## 1 INTRODUCTION

Images captured by different physical sensors under diverse conditions contain unique attributes, which challenge the design of a general image processing system. Drawing on this, to unify the visual pixel distribution at the image level, image fusion provides a solution to combine source images into a single and comprehensive output image [11]. Considering the input configurations, image fusion tasks are ranging from digital image fusion [2, 46], multi-modality image fusion (MMIF) [38], to remote sensing image fusion [12, 40]. In particular, MMIF [19, 49, 50] has attracted wide attention in recent decades, which includes Infrared and Visible Image Fusion (IVIF) [21, 31], and Medical Image Fusion (MIF) [43].

In terms of the IVIF task, the target sources are visible and infrared images [49]. Specifically, the visible modality, captured

by optical devices such as digital cameras, is expert at preserving textural details and colours, while it is susceptible to illumination conditions. On the other hand, the infrared modality, derived from infrared imaging devices, captures the inherent heat radiation emitted by living beings or powered objects. This modality is obtained based on the variation in radiation intensity between the target and its surroundings, lacking the supportive colour and texture contained in the visible spectrum. The IVIF task aims to generate an informative image that leverages the strengths of both, enhancing the overall visual quality and serving various following processing demands [11, 50]. Specifically, the fused images can be applied to several downstream tasks, **e.g.**, semantic segmentation [20, 35], object tracking [4, 13], object detection [8, 34], and saliency detection [27]. As for the MIF task, by combing images obtained through different medical imaging devices (CT [3], MRI, PET, SPETC *e.t.c.*), MIF can provide a clearer view of both the structure and functional information of the human tissues and organs, supporting more precious disease diagnosis [43, 50].

After decades of study, the performance of IVIF has been advanced. In contrast to traditional approaches that rely on multi-scale decomposition (MSD) [52], sparse representation (SR) [45], or low-rank representation (LRR) [14], to extract handcrafted features, more robust representations can be obtained by deep solutions, such as convolutional neural networks (CNNs), Transformers, and their hybrid versions [11]. To pursue better visual effects, various functional blocks have been developed, **e.g.**, aggregated residual dense block [21] and gradient residual dense block [31]. To facilitate smooth training, residual connections [15, 17] and skip connections [37] are adopted to prevent gradient vanishing or distortion in the generated images. Besides modifying network modules, more efforts have been explored to formulate fusion as a generation task. For instance, [19, 23], utilise a generator to obtain the fused image, accompanied by two discriminators to guarantee its fidelity compared to the sources. Essentially, the generator contains multiple CNN layers, and this adversarial training scheme is easy to crash and hard to control. Such computation burden is also suffered by the diffusion solution [50], which obtains stable and controllable high-quality fused images without discriminator but requires more computation resources than previous approaches.

Despite the performance promotion achieved by the above attempts, all the involved network designs suffer from excessive complexity and redundancy. Therefore, striking a balance between model performance and resource requirements is a pressing issue that needs to be addressed. Motivated by knowledge distillation [9, 44], it is straightforward to compress the teacher parameters into a relatively smaller student model. Essentially, we propose to establish a strong teacher model with powerful feature extraction and reconstruction ability. To deliver effective supervision, as shown in Figure 1, we adopt three dedicated designs to integrate supervision signals from teacher, source images, and history records.

Specifically, a digestible distillation strategy is proposed to relax the strict consistency constraints between the student network and the teacher network on the intermediate feature dimension. We construct groups of isomorphic transformation modules within both the teacher and student networks individually. Based on the output characteristics at each stage, we match them from the feature end and output end. To exploit the source input images, intensity

details, edge gradients, and perception semantics are comprehensively reflected by our loss function. Furthermore, to reinforce the cues embedded in the historical parameters of the training process, we propose the dynamic refresh strategy. A set of dual evaluation metrics are devised to distinguish whether the current cues are useful or not. The useful one demonstrates that our training trajectory is correct at present and can be refreshed as internal supervision signals, while the inadequate one can be optimized by the above signals.

In this paper, we design a *Distilled Mini-Model with Dynamic Refresh for Multi-Modality Image Fusion* (MMDRFuse). As shown in Figure 1, our MMDRFuse achieves promising fusion performance, while it is significantly smaller than other SOTA models, with only 113 trainable parameters (0.44 KB). The contributions of our work can be summarised as follows:

- An end-to-end fusion model with a total of 113 trainable parameters and 0.44KB size, which can efficiently facilitate fusion and support downstream tasks. This mini model unveils that boosting image fusion performance benefits more from providing suitable supervision rather than stacking complex networks.
- A digestible distillation strategy to relax the feature-level consistency, softening the supervision from the teacher model.
- A comprehensive loss function to preserve intrinsic clues from source inputs, collaborating pixels, gradients, and perceptions.
- A dynamic refresh strategy to effectively manage the historical states of parameters during training, endowed with a dual evaluation metric system to adaptively refine supervision signals towards the correct direction.
- Promising performance in terms of fusion quality and efficiency against SOTA methods both in IVIF and MIF tasks. The obtained 113 parameters can even support downstream pedestrian detection without any semantic information as input.

## 2 RELATED WORK

### 2.1 Advanced MMIF Formulations

Recent studies in MMIF can be broadly categorised into Auto-Encoder (AE) paradigm [6, 15, 17, 18, 49], GAN-based paradigm [19, 24], Transformer-based paradigm [22, 49], downstream task-driven paradigm [19, 31, 48], and text-driven paradigm [6]. Among the involved techniques, AE exploits an encoder to extract features from source images and a decoder to obtain fused images by reconstructing the latent features. To get rid of the limitations of handcrafted fusion rules, [17] devises a residual fusion network to smooth the training stage. To alleviate computation, [49] uses Lite-Transformer to fuse base features and detail features come from different modalities. Besides network design, [37] directly conducts an end-to-end fusion network with a memory unit to allocate effective supervision signals.

To better support downstream tasks, [19] proposes a bilevel optimisation formulation based on GANs and Object Detection (OD) network, forming a cooperative training scheme to yield optimal network parameters with fast inference for both tasks. Besides, [31]

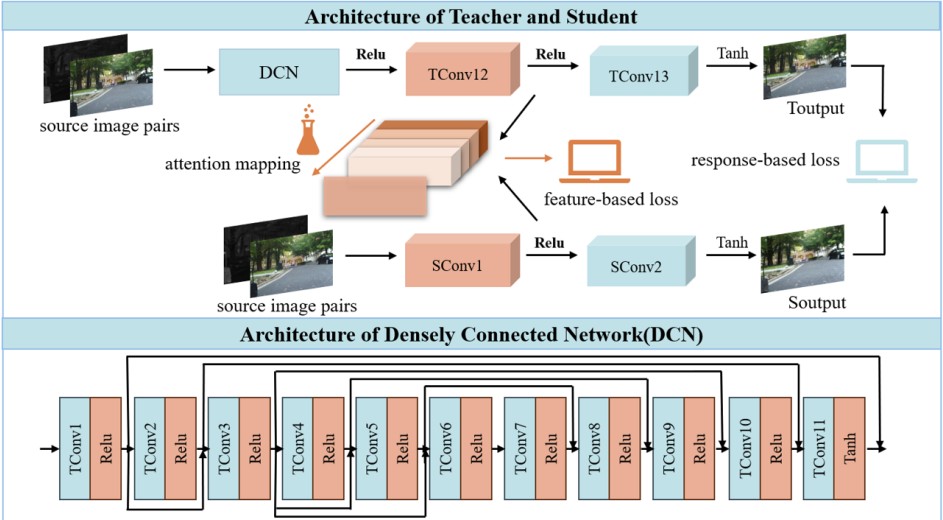

**Figure 2: Illusatration of our distillation process. TConv1, TConv2, ... , TConv13 represent the convolutional layers in the teacher network. SConv1, SConv2 represent the convolutional layers in the student network. TOutput and SOutput denote their outputs respectively.**

introduces a real-time segmentation model to grind the semantic information for the fused images. Although linking the downstream tasks with image fusion can obtain guided semantics, such a solution sacrifices too much in computation, running time, and storage space. To break away from existing modelling techniques, we investigate the possibility of designing internal and external supervision signals to serve an extremely tiny model. In particular, the internal and external supervision denote the historical state of the model itself and the guidance of a powerful teacher model, respectively. Such a combination is typically controlled by a dynamic refresh strategy to harmonise the training of our 0.44KB mini model.

### 2.2 Distillation Techniques

Knowledge Distillation is introduced by [9] to tackle the cumbersome of deploying computationally expensive models. According to the locations of processing knowledge, it can be categorised into three branches, *i.e.*, response-level [9, 47], feature-level [5, 29, 30, 44, 51] and relation-level [25, 42] settings. Typically, [9] distils knowledge by aligning the student's output with the teacher's output while [29] additionally focuses on the consistency of intermediate features between teacher and student. Following this setting, advanced studies are dedicated to paying attention to feature-based distillation. In terms of formulating feature consistency, [44] proves that activation-based attention transfer and gradient-based spatial attention transfer are more effective than full-activation transfer. Considering feature diversity, [5] proposes to gradually learn the low-level feature maps after the high-level consistency is obtained. However, this knowledge delivery solution still costs a lot of resources during the training process.

Given that the structure of our student network is too simple (113 parameters) to directly learn from a powerful teacher, similar to feeding a baby, how to transfer digestible supervision from the teacher to the student poses significant challenges. Drawing on this, we feed the student with more finely processed knowledge

by combining feature-based and response-based distillation within isomorphic transformation modules. This process can also be understood as a mother carefully preparing swallowable pureed food for her child.

## 3 APPROACH: MMDRFUSE

### 3.1 Digestible Distillation

In essence, the ultimate goal of our MMDRFuse is to obtain a mini model with satisfactory fusion performance and downstream supportive power. It is impossible to completely depend on directly training the mini model itself, which is greatly limited by its parameter volume. Accordingly, we first train a teacher model, which is utilized to learn considerable feature extraction and reconstruction ability. Furthermore, the key of teacher module is to as a transportation media, converting high-level abstract feature information into fragments that student can digest.

Existing observation unveils that aligning knowledge at the output end is too violent for the student model. In particular, different from high-level visual tracks, such as classification and detection, the output of a fusion system is low-level pixels, which challenges the strictness of teacher guidance. Therefore, we propose to add a buffer before the final output to transfer the supervision that can be digested better. At the same time, feature maps of the middle layer is supposed to be consistently distributed as the teacher model [44], which can help to produce the expected output.

We adopt a relatively complex teacher network to obtain a model with robust fusion ability. This network incorporates a densely connected network (DCN) responsible for extracting deep image features comprising 64 channels. Additionally, it integrates two convolutional layers to generate four-channel feature maps, alongside a single-channel image. By contrast, our student model only comprises two convolutional layers. The first one is used to extract feature maps with four channels and the latter functions as a

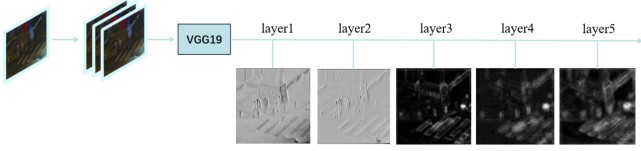

**Figure 3: Illustration of the feature maps used to reflect perception degrees. From left to right, it represents the source image, duplicated source image, and five feature maps extracted by VGG-19, respectively.**

decoder for producing the fused image. We utilise each output of the student network to implement feature-based and response-based distillation with the last two outputs of the teacher, respectively. Figure 2 provides an illustration of the network architecture and distillation mechanism.

To ensure that the student outputs closely resemble those of the teacher network, we encourage consistency between their outputs at the pixel level. Specifically, the feature-based distillation and the response-based distillation can be formulated as:

$$L_{distill} = \frac{1}{2} \sum_{i=1}^{2} \| \frac{vec(F(T_i))}{\|vec(F(T_i))\|_2} - \frac{vec(F(S_i))}{\|vec(F(S_i))\|_2} \|_2, \quad (1)$$

where $vec(\cdot)$ denotes the vectorisation operation. $F(x) = \sum_{i=1}^{C} x_i$ is a spatial mapping function used to conduct attention mapping across channel dimensions. $F(x)$ comprehensively considers all channels from $x$ by distributing average weights to each spatial region. On the other hand, for the feature-based distillation, $S_1 \in R^{H \times W \times 4}$ and $T_1 \in R^{H \times W \times 4}$ are the extracted features of the teacher network and student network, respectively. For the response-based distillation, $S_2 \in R^{H \times W \times 1}$ and $T_2 \in R^{H \times W \times 1}$ are the output images of these two networks. As these output feature maps and images share an identical number of channels, the teacher network can effectively impart knowledge to the student module at the same semantic level.

## 3.2 Comprehensive Loss

In this section, we introduce our loss function for optimising the fusion network from the source inputs. Firstly, we expect the outputs to be similar to source images at the pixel level. Specifically, for the multimodal image fusion task, visible or MRI images usually contain information about the light condition with texture details, while infrared or PET images tend to present significant thermal and functional information. Thus, an intensity loss function [31] is designed to generate fused images that share the same pixel distribution with source inputs, *i.e.*

$$L_{int} = \frac{1}{HW} \|O - max(I_{ir}, I_{vis})\|_1, \quad (2)$$

where $I_{ir}$ and $I_{vis}$ denote the source infrared and visible images, $O$ represents the output of the fusion network. Notably, if source images are under normal light conditions, the intensity loss function can help to maintain detailed targets, clear structure, *etc.* However, if the light condition in texture areas is darker than the infrared image, the impact is dominated by the latter. Thus, it seems not enough to rely solely on the intensity loss function. This phenomenon can be observed by the examples of Figure 5. Therefore, we add a maximum

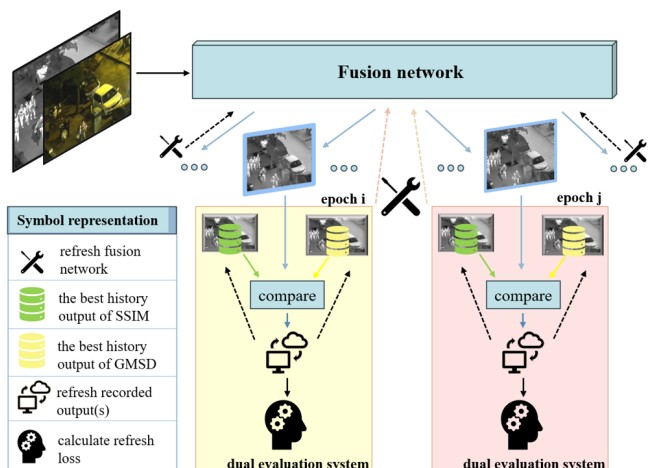

**Figure 4: Workflow of the proposed dynamic refresh strategy.**

gradient loss to help transfer gradient information mainly from the visible image, it is defined as:

$$L_{grad} = \frac{1}{HW} \| \nabla O - max(\nabla I_{ir}, \nabla I_{vis})\|_F^2, \quad (3)$$

where $\| \cdot \|_F$ represents the Frobenius norm, $\nabla$ represents gradient operator. Furthermore, to better fuse fine-grained features from source images, we employ a normalised VGG-19 network [21] to extract hierarchical features of input and output images, formulating a feature-based perception loss. As depicted in Figure 3, the feature maps in the shallow layers exhibit clearer edge texture details and gradient information, while the feature maps in deep layers become blurred, encompassing higher-level semantic information, abstract contexts, and salient structure information. This information is a significant factor regarding the fusion performance [37]. Hence, the perception loss is formulated as:

$$L_{percep} = \frac{\sum_{i=1}^{5} \sum_{j=1}^{D_i} \|\phi_{i,j}(O) - max(\phi_{i,j}(I_{ir}), \phi_{i,j}(I_{vis}))\|_F^2}{5 \cdot H \cdot W \cdot D_i}, \quad (4)$$

where $D_i$ represents the number of channels of extraction layer $i$, $\phi_{i,j}$ represents $j-th$ channel of accordingly $i-th$ feature map. Finally, the above components collectively constitute the comprehensive loss function:

$$L_{comp} = \gamma L_{int} + \delta L_{grad} + L_{percep}. \quad (5)$$

By adding the maximum gradient loss and maximum perception loss, our fusion network not only focuses on regions with high brightness but also pays attention to texture information in dark areas, which cannot be achieved solely by intensity loss. At the same time, without intensity loss, we can not retain salient targets only by gradient loss and perception loss. Hence, the combination of these loss functions can complement each other's weaknesses while keeping their respective strengths.

## 3.3 Dynamic Refresh Strategy

Traditional methods often directly discard the states of parameters from intermediate iterations during the training process, which, we argue, can be properly utilised to serve as supervision signals.

However, useful information can not be guaranteed in the intermediate iterations, sometimes including noises, artefacts and so on. Hence, we need to identify the effectiveness of them. We devise a dual evaluation system to identify them and further utilise the advantageous parameters. We call this process as dynamic refresh, Figure 4 illustrates the basic process. In particular, dynamic refresh is applied throughout the entire training process, and we adopt two metrics to measure whether it is helpful to fusion. Image quality assessment structural similarity (SSIM) [36] and Gradient Magnitude Similarity Deviation (GMSD) [41] are involved as the metrics. SSIM measures the previous iterations from the perspective of structure, illuminance, and contrast, while GMSD highlights texture details. Accordingly, we measure every output during the training process and keep the two best outputs according to two evaluation metrics, which correspond to the green database and the yellow database in Figure 4, respectively. The evaluation process can be described as follows:

$$\begin{cases} S_{bs} = SSIM(O_{bs}, I_{ir}) + SSIM(O_{bs}, I_{vis}) \\ S_{cur} = SSIM(O_{cur}, I_{ir}) + SSIM(O_{cur}, I_{vis}) \end{cases}, \quad (6)$$

$$\begin{cases} G_{bg} = GMSD(O_{bg}, I_{ir}) + GMSD(O_{bg}, I_{vis}) \\ G_{cur} = GMSD(O_{cur}, I_{ir}) + GMSD(O_{cur}, I_{vis}) \end{cases}, \quad (7)$$

where $O_{cur}$ symbols the output of current epoch, $O_{bs}$ symbols the historical output with the best SSIM value, $O_{bg}$ symbols the historical output with the best GMSD value. $S_{bs}, S_{cur}$ represent the SSIM value of the best history output and the current output, respectively. $G_{bg}, G_{cur}$ represent the GMSD value of the best history output and the current output, respectively.

By calculating SSIM and GMSD values of the current output and the two best historic outputs, once the value of the current output is higher than the recorded outputs, they will be replaced by the current output, which achieves dynamic update. The refreshing process can be described as follows:

$$O_{bs} = \begin{cases} O_{bs}, & \text{if } S_{bs} \geq S_{cur} \\ O_{cur}, & \text{if } S_{bs} < S_{cur} \end{cases}, \quad O_{bg} = \begin{cases} O_{bg}, & \text{if } G_{bg} \geq G_{cur} \\ O_{cur}, & \text{if } G_{bg} < G_{cur} \end{cases}. \quad (8)$$

On the contrary, the current output can learn from historic outputs by following refresh loss:

$$L_s = \frac{\sum_{i=4}^{5} \sum_{j=1}^{D_i} \|\phi_{i,j}(O_{cur}) - \phi_{i,j}(O_{bs})\|_F^2}{2 \cdot H \cdot W \cdot D_i} + \frac{\|O_{cur} - O_{bs}\|_1}{HW} \quad (9)$$

$$L_g = \frac{\sum_{i=1}^{3} \sum_{j=1}^{D_i} \|\phi_{i,j}(O_{cur}) - \phi_{i,j}(O_{bg})\|_F^2}{3 \cdot H \cdot W \cdot D_i} + \frac{\|\nabla O_{cur} - \nabla O_{bg}\|_F^2}{HW} \quad (10)$$

Motivated by the phenomenon presented in Figure 3, we combine perception loss of the last two feature maps before max-pooling operation with intensity loss function, to collectively serve as supervision signals by the item of $L_s$. Similarly, we combine perception loss of the first three feature maps before the max-pooling operation with maximum gradient loss function, to collectively serve as supervision signals by the item of $L_g$. $L_s$ can help to fuse context and spatial structure, brightness, and contrast information from $O_{bs}$, while $L_g$ can help to fuse texture detail and gradient information from $O_{bg}$. Furthermore, we measure the gap between $S_{bs}$ and $S_{cur}$,

as well as $G_{bg}$ and $G_{cur}$ to serve as two self-adaptive coefficient:

$$\begin{cases} gap_s = S_{cur} - S_{bs} \\ gap_g = G_{cur} - G_{bg} \end{cases}. \quad (11)$$

Hence, our dynamic refresh loss can be described as follows:

$$L_{refresh} = gap_s L_s + gap_g L_g. \quad (12)$$

Finally, we use comprehensive loss and refresh loss to collectively train our teacher model and further combined with distillation loss to train our student model. The total loss function can be described as follows:

$$L_{total} = \theta L_{distill} + \lambda L_{comp} + L_{refresh}. \quad (13)$$

## 4 EVALUATION

### 4.1 Setup

*4.1.1 Datasets and Metrics.* We perform MMDRFuse on three types of fusion tasks: Infrared and Visible Image Fusion (IVIF), Medical Image Fusion (MIF), and Pedestrian Detection (PD). We select six SOTA methods to compare with our method, including TextFusion [6], MUFusion [37], LRRNet [16], MetaFusion [48], DeFusion [18], and TarDAL [19]. To better prove the generalisation ability of our design, we only train our model (both teacher and student) on the IVIF task, and directly apply the trained model to other tasks. The training dataset is MSRS [32], we select 1083 image pairs from it, and they are cropped into 128×128 image patches, ending with 16245 image pairs. For the MMIF task, we conduct test experiments on three datasets: MSRS, LLVIP [10], and RoadScene [39]. The image numbers of them are 361, 250, and 50, respectively. For the MIF task, we directly test the model on MRI-SPECT and MRI-PET, which include 73 and 42 image pairs, respectively.

For the PD task, we trained six SOTA methods and our method on 2000 image pairs from the MSRS dataset, and then test the detection effect on 250 image pairs from the LLVIP dataset. We perform PD experiments by using the Yolov5 as a detector to evaluate the pedestrian detection performance with the value of mAP@.5:.95, which provides a more comprehensive evaluation by calculating the average precision according to the IOU threshold from 0.5 to 0.95, taking into account the performance of the model throughout the entire retrieval process. The training epoch, batch size, and optimizer are set as 3, 16, and SGD optimiser, respectively.

Besides, we adopt six metrics to fairly judge our method including entropy (EN) [1], standard deviation (SD) [33], mutual information (MI) [28], visual information fidelity (VIF) [7], structural similarity index measure (SSIM) [36], and $Q^{AB/F}$ [26]. EN measures fused images from the perspective of information theory, which is primarily used to assess the amount of information and complexity contained in the fused image. SD is a statistical metric, that measures an image from its brightness and contrast, but sometimes it can be disturbed by noise. MI measures the amount of information that two images share in common. VIF not only considers the pixel-level similarity with source images, but also cares about human visual perception. SSIM measures fused images from structure, illuminance and contrast. $Q^{AB/F}$ not only focuses on the visual quality of the image but also takes into account the preservation of the image content and structure.

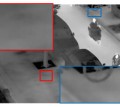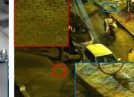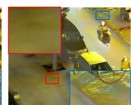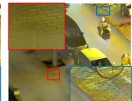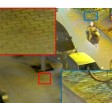

**Infrared image**   **Visible image**   **Stage I**   **Stage II**   **Stage III**

**Figure 5: The visualised validity verification of the comprehensive loss function and dynamic refresh retrain strategy. Stage I: only using intensity loss function, Stage II: adding gradient loss function and perception loss function, Stage III: combining with dynamic refresh strategy.**

*4.1.2 Implement Details.* Our experiments are conducted on a Linux server with an NVIDIA GeForce RTX 3090 GPU. We first train the teacher model with 8 epochs and the number of batch size is 45. Then we adopt the trained teacher models to collectively train the student model with 30 epochs and the batch size is 30. For hyperparameters setting, we set $\gamma = 1$, $\delta = 0.1$, $\theta = 0$ and $\lambda = 1$ for teacher model, and $\gamma = 2$, $\delta = 1$, $\theta = 0.1$ and $\lambda = 1$ for student model, respectively. In terms of the optimiser, we utilise the Adam optimizer with an initial learning rate equal to $10^{-4}$.

## 4.2 Ablation Studies

Due to the dynamic refresh being impacted by the comprehensive loss items, we first verify the efficacy of comprehensive loss, including intensity loss, maximum gradient loss, and maximum perception loss, individually. Subsequently, we demonstrate the effectiveness of our dynamic refresh strategy and the distillation mechanism step by step.

*4.2.1 Comprehensive Loss Function and Dynamic Refresh Strategy.* As we mentioned in the comprehensive loss function part, the intensity loss is mainly used to retain the pixel distribution, salient targets and general structural information. The maximum gradient item and maximum perception item are further utilised to retain texture, gradient and high-level semantic information. We initially explore their effects on the teacher model. From Figure 5, we can see the fused image only trained by intensity loss (Stage I), for areas with lower brightness in visible light images like the blue box and red box, it indeed fails to preserve textural details and gradient information. After applying the maximum gradient loss and maximum perception loss function (Stage II), from the ground shadows within the red and blue boxes, it can be observed that the missing information becomes increasingly rich and clearly visible.

Our dynamic refresh strategy is mainly adopted to sufficiently explore the vital information contained in intermediate outputs. With the combined effect of comprehensive loss functions, the extraction power becomes increasingly strong, fully absorbing the information contained in not only the source images but also the historical reference outputs. From the last image in Figure 5 (Stage III), We can see the information within the red and blue boxes displayed with unprecedented clarity, little artifact and noise. Besides, quantitative results displayed in Table 1 show that our comprehensive loss function and dynamic refresh strategy are effective.

**Table 1: Quantitative ablation study results of comprehensive loss and dynamic refresh on LLVIP. Boldface and underline show the best and second-best values, respectively.**

| | Configurations | SD | VIF | $Q^{AB/F}$ | SSIM |
|---|---|---|---|---|---|
| I | Before $L_{grad}$ & $L_{percep}$ | 48.40 | 0.83 | 0.51 | 0.85 |
| II | After $L_{grad}$ & $L_{percep}$ | 48.26 | 0.87 | 0.61 | 0.87 |
| III | After dynamic refresh | **49.39** | **0.94** | **0.70** | **0.90** |

**Table 2: Quantitative ablation study results of distillation on LLVIP. Boldface shows the best value.**

| | Configurations | SD | VIF | $Q^{AB/F}$ | SSIM |
|---|---|---|---|---|---|
| I | without distillation | 44.78 | 0.73 | 0.48 | 0.79 |
| II | with direct distillation | 44.93 | 0.77 | 0.56 | 0.78 |
| III | with digestible distillation | **46.98** | **0.81** | **0.57** | **0.85** |

**Table 3: Quantitative ablation student model size of distillation on LLVIP. Boldface shows the best value.**

| | Model Size(kb) | SD | VIF | $Q^{AB/F}$ | SSIM |
|---|---|---|---|---|---|
| student1 | 1021.01 | 49.80 | 0.81 | 0.62 | 0.87 |
| student2 | 7.01 | **51.07** | 0.81 | 0.63 | 0.89 |
| student3 | **0.44** | 46.98 | 0.81 | 0.57 | 0.85 |
| teacher | 4347.57 | 49.39 | **0.94** | **0.70** | **0.90** |

*4.2.2 Distillation Mechanism.* We design a student model with only two convolutional layers and 113 trainable parameters by utilising a powerful teacher. Specifically, distillation is performed at the feature and response ends. To verify the effectiveness of distillation, we conduct an ablation study including the student model without distillation, the student model with direct distillation, and the student model with our digestible distillation. From the first two rows on Table 2, we can observe that students can only learn a little bit of the superficial aspects from the teacher through direct distillation, resulting in a decrease in SSIM. In contrast, through our digestible distillation, all four metrics have shown notable improvements. Positive evidence of these four metrics means the quality of the fused image itself is higher than before, sharing more detailed information with source images from both pixel level and visual perception perspectives.

In fact, there are many volumes of targeted students available for us to choose from. During this process, we gradually compress the student model down to the minimum, where different model sizes correspond to different compression ways and varying effects. As displayed in Table 3, from the perspective of metrics value, we should opt for the second student model. However, in pursuit of an extremely mini model size, we choose the third configuration, which achieves an extremely tiny size, while dose not compromising performance to a significant extent.

Finally, we obtain the mini model with only 113 trainable parameters. Before we display the fusion performance of our experiments, we first compare the model efficiency and complexity with several SOTA approaches, as displayed in Table 4. Among all the involved methods, our MMDRFuse exhibits the smallest model size, the least number of floating-point operations, and the fastest average running time. In the following experiments, we will further demonstrate that our MMDRFuse not only enjoys advantages in efficiency and complexity but also leads to promising model performance.

**Table 4: Comparison of Model Efficiency and Complexity with SOTA approaches. Boldface shows the best value.**

| Methods | Venue | Size(KB) | Flops(G) | Time(MS) |
|---|---|---|---|---|
| TextFusion | 24' Arxiv | 288.51 | 88.708 | 11.17 |
| MUFusion | 23' Inf. Fus. | 4333.76 | 240.669 | 7.91 |
| LRRNet | 23' TPAMI | 192.20 | 60.445 | 523.35 |
| MetaFusion | 23' CVPR | 3170.76 | 1063.000 | 325.37 |
| DeFusion | 22' ECCV | 30759.66 | 322.696 | 339.59 |
| TarDAL | 22' CVPR | 1158.50 | 388.854 | 124.10 |
| MMDRFuse | Ours | **0.44** | **0.142** | **0.62** |

## 4.3 Infrared and Visible Image Fusion

*4.3.1 Qualitative Comparison.* We show the qualitative comparison results in Figure 6 and Figure 7. In Figure 6, MUFusion, LRRNet, and MetaFusion not only distort the colour of the sky but also exhibit numerous artifacts around the license plate, as shown within the yellow box. Besides, DeFusion and TextFusion result in a blurry appearance of the leaves within the red box. TarDAL shows a relatively apparent contrast in the red box, but the colour of the building seems so bright that we can not clearly see the content within it. While our MMDRFuse can show a clear result in the above points. Furthermore, Our method can illuminate objects in the dark, even though those objects are not very clear in the source images, such as the pillar in Figure 7.

*4.3.2 Quantitative Comparison.* To objectively demonstrate the performance of MMDRFuse, we conduct quantitative experiments on three typical datasets, which include MSRS, LLVIP, and Road-Scene. The MSRS dataset covers scenes from urban streets to rural roads under various lighting conditions, containing rich semantic information. The LLVIP dataset mainly includes urban and street environments under low-light conditions. The RoadScene dataset, on the other hand, contains scenes with roads, vehicles, pedestrians, and more under various lighting conditions. We show the quantitative results in Table 5. Our method consistently ranks first in three metrics (MI, VIF, $Q^{AB/F}$) across the three datasets, and first or second in two metrics (EN, SSIM). Besides, in terms of the SD metric that is susceptible to noise, we rank first on MSRS, second on RoadScene, and do not perform particularly well on LLVIP, which is a common occurrence and does not detract from our excellent performance.

The outstanding performance in full-reference (MI, VIF, $Q^{AB/F}$, SSIM) and no-reference (EN, SD) indicators show that the fusion results not only closely resemble the source images at the visual and pixel levels, but also possess rich, high-quality information within each fused image itself. The above results also demonstrate that our mini model can adeptly handle a variety of scenes and lighting conditions, and it surpasses much larger models in both visual effects and evaluation metrics, perfectly achieving a balance between performance and various costs.

## 4.4 Medical Image Fusion

*4.4.1 Qualitative Comparison.* To verify the generalisation ability of our method, we directly apply the above six SOTA methods and our design to MIF without retraining. We show the qualitative results of MRI-PET images in Figure 8. From the green and red boxes, we can observe that our MMDRFuse not only preserves the detailed internal structural information displayed in MRI images

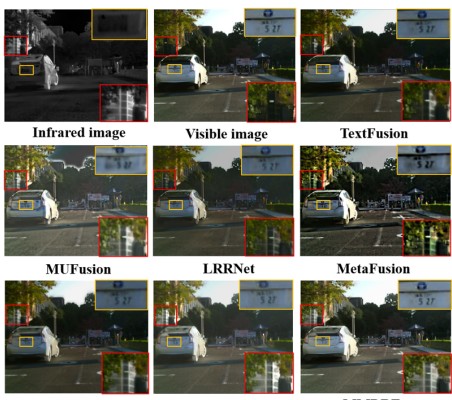

**Figure 6: Visual comparison with SOTA on MSRS.**

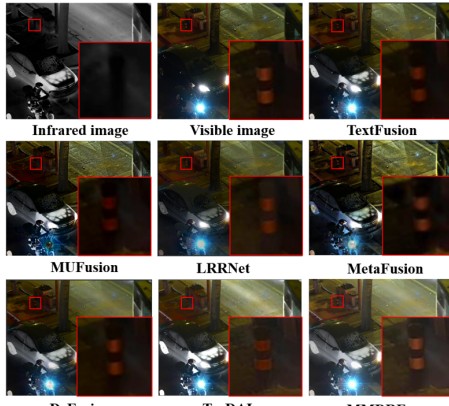

**Figure 7: Visual comparison with SOTA on LLVIP.**

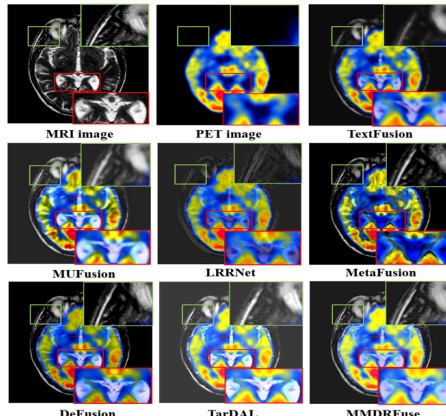

**Figure 8: Visual comparison with SOTA on MRI-PET.**

but also retains the distribution of the radioactive tracers shown in PET images, providing a more comprehensive set of diagnostic information. While TextFusion, LRRNet, and MetaFusion lose the information of internal structure. MUFusion, LRRNet, MetaFusion, and DeFusion dilute the distribution of the radioactive tracers. MU-Fusion and TarDAL introduce extra artifacts in the detailed internal structure.

**Table 5: Quantitative results of the IVIF task. Boldface and underline show the best and second-best values, respectively.**

**(a) Dataset: MSRS Infrared-Visible Dataset**

| Methods | EN | SD | MI | VIF | $Q^{AB/F}$ | SSIM |
|---|---|---|---|---|---|---|
| TextFusion | 6.03 | 38.02 | 2.44 | 0.72 | 0.52 | 0.76 |
| MUFusion | 5.96 | 28.48 | 1.17 | 0.60 | 0.42 | 0.71 |
| LRRNet | 6.19 | 31.76 | 2.03 | 0.54 | 0.45 | 0.43 |
| MetaFusion | 6.37 | 39.64 | 1.16 | 0.71 | 0.48 | 0.78 |
| DeFusion | 6.34 | 34.86 | 2.15 | 0.75 | 0.51 | 0.93 |
| TarDAL | 6.48 | 37.58 | 1.82 | 0.70 | 0.42 | 0.70 |
| MMDRFuse | **6.81** | **45.37** | **3.02** | **1.00** | **0.64** | **1.00** |

**(b) Dataset: LLVIP Infrared-Visible Dataset**

| Methods | EN | SD | MI | VIF | $Q^{AB/F}$ | SSIM |
|---|---|---|---|---|---|---|
| TextFusion | 7.05 | 47.82 | 1.99 | 0.71 | 0.53 | 0.77 |
| MUFusion | 7.03 | 40.44 | 1.69 | 0.68 | 0.47 | 0.70 |
| LRRNet | 6.67 | 35.42 | 1.64 | 0.56 | 0.47 | 0.65 |
| MetaFusion | 7.14 | 48.90 | 1.16 | 0.61 | 0.29 | 0.60 |
| DeFusion | 7.25 | 43.93 | 2.33 | 0.74 | 0.43 | 0.83 |
| TarDAL | **7.58** | **61.77** | 2.21 | 0.65 | 0.42 | 0.71 |
| MMDRFuse | 7.34 | 46.98 | **3.02** | **0.81** | **0.57** | **0.85** |

**(c) Dataset: RoadScene Infrared-Visible Dataset**

| Methods | EN | SD | MI | VIF | $Q^{AB/F}$ | SSIM |
|---|---|---|---|---|---|---|
| TextFusion | 6.86 | 38.43 | 2.45 | 0.68 | 0.44 | **0.94** |
| MUFusion | **7.51** | **54.92** | 1.72 | 0.51 | 0.34 | 0.78 |
| LRRNet | 7.12 | 43.90 | 2.09 | 0.51 | 0.38 | 0.71 |
| MetaFusion | 7.17 | 48.66 | 1.60 | 0.51 | 0.36 | 0.75 |
| DeFusion | 6.80 | 32.86 | 2.08 | 0.50 | 0.38 | 0.86 |
| TarDAL | 7.21 | 44.88 | 2.43 | 0.53 | 0.39 | 0.83 |
| MMDRFuse | 7.24 | 54.45 | **3.09** | **0.72** | **0.45** | 0.89 |

*4.4.2 Quantitative Comparison.* We display the quantitative results on both MRI-SPECT and MRI-PET image pairs in Table 6. Our MMDRFuse ranks first in four metrics (SD, MI, VIF, $Q^{AB/F}$) and ranks second or third in two metrics (EN, SSIM). Combined with the above qualitative results we can notice that methods (MUFusion, TarDAL) introduce extra artifacts that can reach high values of EN, methods (MetaFusion, DeFusion) dilute the distribution of the radioactive tracers can obtain high values of SSIM, which can not provide much assistance for diagnosis. Therefore, after performing a comprehensive analysis of the above results, our MMDRFuse with an extreme mode size and considerable performance is better suited to meet the needs of medical image fusion.

## 4.5 Pedestrian Detection

To further validate the performance of our MMDRFuse on downstream detection tasks, we conduct experimental exploration on pedestrian detection. As can be seen from Figure 9, although LRR-Net ([16]) achieves the best detection precision, it requires significantly more time and computational resources than our design. Furthermore, compared with MetaFuse ([48]), which introduces the detection task as additional supervision, our method performs better even without any prior settings in detecting pedestrians. This can be attributed to the introduction of perceptual loss, combined with the dynamic refresh strategy and distillation mechanism that jointly learns advanced semantic information. In contrast, our method can precisely detect pedestrians from significant targets

**Table 6: Quantitative results of the MIF task. Boldface and underline show the best and second-best values, respectively.**

**(a) Dataset: MRI-SPECT Infrared-Visible Dataset**

| Methods | EN | SD | MI | VIF | $Q^{AB/F}$ | SSIM |
|---|---|---|---|---|---|---|
| TextFusion | 3.83 | 41.32 | 1.77 | 0.52 | 0.21 | 0.32 |
| MUFusion | 4.37 | 54.19 | 1.65 | 0.46 | 0.44 | 0.36 |
| LRRNet | 3.98 | 42.16 | 1.63 | 0.34 | 0.20 | 0.21 |
| MetaFusion | 3.62 | 45.43 | 1.64 | 0.47 | 0.39 | 1.40 |
| DeFusion | 3.73 | 51.33 | 1.82 | 0.59 | 0.55 | **1.47** |
| TarDAL | **4.69** | 58.41 | 1.79 | 0.56 | 0.48 | 0.37 |
| MMDRFuse | 3.99 | **64.11** | **1.96** | **0.61** | **0.63** | 0.40 |

**(b) Dataset: MRI-PET Infrared-Visible Dataset**

| Methods | EN | SD | MI | VIF | $Q^{AB/F}$ | SSIM |
|---|---|---|---|---|---|---|
| TextFusion | 4.30 | 60.1 | 1.74 | 0.63 | 0.36 | 0.38 |
| MUFusion | **4.82** | 60.81 | 1.54 | 0.43 | 0.42 | 0.38 |
| LRRNet | 4.36 | 48.30 | 1.56 | 0.37 | 0.21 | 0.22 |
| MetaFusion | 4.08 | 63.47 | 1.68 | 0.50 | 0.51 | **1.43** |
| DeFusion | 3.32 | 49.26 | 1.40 | 0.45 | 0.45 | 1.18 |
| TarDAL | 4.38 | 52.99 | 1.79 | 0.46 | 0.45 | 0.30 |
| MMDRFuse | 4.38 | **75.22** | **1.93** | **0.65** | **0.69** | 0.45 |

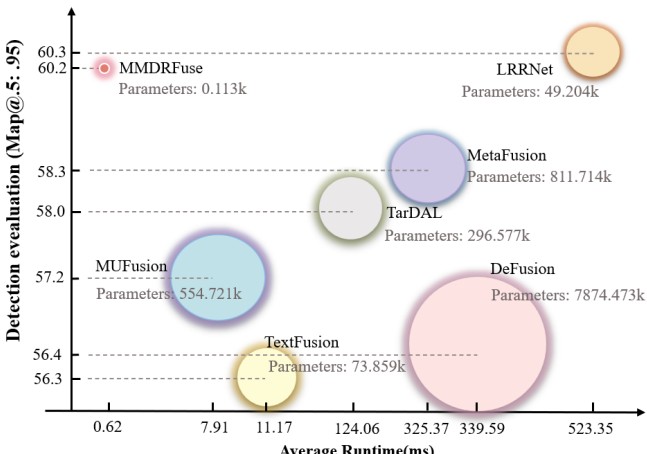

**Figure 9: Performance comparison in terms of average detection precision and average speed on RGBT pedestrian detection.**

within contrasting backgrounds without detecting semantic information as input and with less running time as well as a mini model size under 1 KB, achieving satisfactory detection results at a considerably marginal cost.

## 5 CONCLUSION

An investigation into compressing the image fusion model is conducted in this work. We utilise the specially designed comprehensive loss function and the dynamic refresh strategy based on intermediate fusion results to first formulate a fusion model. Further combined with the digestible distillation strategy, we successfully train an extremely tiny (0.44 KB) student network from the teacher model. Experimental results on multiple tasks demonstrate that our mini model not only exhibits advantages in efficiency and complexity but also achieves promising results on the MMIF and downstream detection tasks.

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
