# OpenReview forum: "MMDRFuse: Distilled Mini-Model with Dynamic Refresh for Multi-Modality Image Fusion"
_acmmm.org/ACMMM/2024/Conference — MM2024 Oral_

### Official Review · Reviewer_vg2R · 2024-05-19

**Rating:** 5
**Confidence:** 3

**Summary:**

This paper proposes a lightweight Distilled Mini-Model, named MMDRFuse, for multi-modality image fusion. The MMDRFuse has only 113 parameters (0.44 KB) but achieves a good performance as well as other bigger models, which is achieved by three carefully designed supervisions. The first is a two-stage distillation, which aims to distill knowledge from the feature and output. The second is the Comprehensive Loss, which involves image-level and feature-level complementary. The third is a Dynamic Refresh Strategy, which applies Image quality as an assessment structural similarity (SSIM) and Gradient Magnitude Similarity Deviation (GMSD) as the metrics to measure whether the current model is the best one. Experiments are conducted on several public benchmark datasets and a pedestrian detection task.

**Strengths:**

The method is novelty, and the writing is good and easy to follow.

**Limitations:**

1.	The granularity of ablation experiments should be more detailed, e.g., more ablation experiments should be performed where any one of the three items in the 𝐿_{𝑐𝑜𝑚𝑝} loss is missing.
2.	Figure 4 is confusing, more notations and comments should be added.
3.

**Suitability:**

3

---

### Official Review · Reviewer_yTbQ · 2024-05-25

**Rating:** 3
**Confidence:** 4

**Summary:**

The paper primarily discusses the training of a minimal student model using distillation, a comprehensive loss function, and dynamic refresh strategies. It provides a detailed explanation of the Digestible Distillation technology, comprehensive loss function, and Dynamic Refresh Strategy techniques used in this paper. In terms of experiments, the model is trained on Visible Image Fusion (IVIF), Medical Image Fusion (MIF), and Pedestrian Detection (PD), achieving generalization from the IVIF task to the other two tasks. Ablation studies are conducted to compare the effects of distillation with and without distillation, direct distillation, and the new distillation method proposed in this paper. Comparisons are also made regarding the use of the comprehensive loss function and the Dynamic Refresh Strategy. Finally, experimental results are presented to show the performance of the IVIF, MIF, and PD tasks in comparison with other state-of-the-art (SOTA) methods.

**Strengths:**

Directly transferring knowledge from the teacher model to the student model is unstable, so the authors propose adding a buffer before the final output to enable the student model to digest it better. The teacher model uses a densely connected network (DCN) to extract image features, which is then followed by two convolutions to produce feature maps with 4 channels. To achieve lightness, the student model only adopts two convolution operations: the first convolution is used to extract feature maps from the teacher model, and the second convolution acts as a decoder to produce the fused image. The Comprehensive Loss introduces an intensity loss to ensure that the fused image has the same pixel distribution as the input images, a gradient loss is sued to help visible images transfer gradient information, and a perception loss is mainly used to eliminate the blurring of feature maps and abstract information in the fusion performance. The dynamic refresh strategy, where parameter changes during the training process are often neglected in traditional methods, utilizes these changes as supervisory signals. Sometimes supervisory signals may contain some useless information, such as noise, so an evaluation system is designed to determine usefulness, calculating SSIM and GMSD for intermediate results and taking the optimal values.

**Limitations:**

Achieving good performance across multiple task metrics with a small model is commendable, but the distillation strategy lacks innovation. Moreover, we should focus more on unsupervised or weakly supervised approaches, as the cost of obtaining supervised signals is relatively high. No ablation experiments are conducted on the three losses separately within the comprehensive Loss, so it is uncertain which loss truly has the effect. The details and the purpose of the dynamic refresh strategy are not clear, how should we interpret "to strengthen the cues embedded in the historical parameters of the training process, we propose the dynamic refresh strategy"? Although knowledge distillation based method proposed in this paper works well, it is not very innovative approaches in idea.
In addition, the SD in Figure 1 and the SSIM in the relevant description make me confused.

**Suitability:**

2

---

### Official Review · Reviewer_AvCz · 2024-05-30

**Rating:** 5
**Confidence:** 2

**Summary:**

They utilise the specially designed comprehensive loss function and the dynamic refresh strategy based on intermediate fusion results to first formulate a fusion model. They successfully train an extremely tiny (0.44 KB) student network from the teacher model with the smallest model size, the fastest running time and the best performance.

**Strengths:**

Extensive experiments on several public datasets demonstrate that their method exhibits promising advantages in terms of model efficiency and complexity, with superior performance in multiple image fusion tasks and downstream pedestrian detection application, with the smallest model size, the fastest running time and the best performance.

**Limitations:**

Based on my limited expertise in the field, I find the paper to be impressive. The extensive experiments conducted on various public datasets demonstrate the method's promising advantages in terms of model efficiency, complexity, and performance in image fusion tasks and pedestrian detection applications. The authors' focus on the smallest model size, fastest running time, and superior overall performance adds significant value to the research field. Therefore, my rating is Weak Accept.

**Suitability:**

2

---

### Official Review · Reviewer_2Hzj · 2024-05-31

**Rating:** 5
**Confidence:** 3

**Summary:**

This paper proposes a well-designed distillation strategy to transfer the knowledge from powerful teacher model to extremely small student model.
The authors adopt three dedicated designs to integrate supervision signals from teacher, source images, and history records.
Experiments demonstrate the efficiency and effectiveness of the proposed method, with superior performance in multiple image fusion tasks and downstream pedestrian detection application.

**Strengths:**

1. This paper proposes an end-to-end extremely small fusion model with a total of 113 trainable parameters (0.44 KB) and promising advantages in terms of model efficiency and complexity, it greatly promots the development of multiple image fusion tasks.
2. The dynamic refresh training strategy identifies the useful information in the intermediate iterations, and refine the supervision signals adaptively. This method is innovative and proven effective in Table 1.
3. Efficiency and effectiveness of the proposed method is validated in multiple image fusion tasks and downstream pedestrian detection application.

**Limitations:**

1. In Table 1, the performance of student1 is worse than that of student2, while the model size of student1 is 2 orders of magnitude larger than that of student2. The results look a bit strange, I'm not sure whether student1 is trained enough.
2. The third row in Table 2 appears to use a combination of digestible distillation loss, comprehensive loss, and dynamic refresh loss. Because the quantitative result in Table 2 is the same as that in Table 3 and Table 5 (b).
3. It lacks details about the average running time in Table 4, how do you get the time ?
4. Will you plan to release the code ?

**Suitability:**

3

---

### Meta-Review · Area_Chair_YyrW · 2024-06-29

**Recommendation:** Accept (Oral)
**Confidence:** 5

**Metareview:**

This work proposes a lightweight Distilled Mini-Model with a Dynamic Refresh strategy (MMDRFuse) for Multi-Modality Image Fusion, achieving model parsimony through careful supervisions and demonstrating advantages in experiments and downstream applications. With an average score of 5 and the unanimous positive reviews, the paper is competitive for this year's ACM MM conference. The AC agrees to accept the paper and strongly recommends incorporating the content from the rebuttal into the final version.